# TabMDA: Tabular Manifold Data Augmentation for Any Classifier using Transformers with In-context Subsetting

**Andrei Margeloiu** [* 1]  **Adrián Bazaga** [* 1]  **Nikola Simidjievski** [2 1]  **Pietro Liò** [1]  **Mateja Jamnik** [1]

## Abstract

Tabular data is prevalent in many critical domains, yet it is often challenging to acquire in large quantities. This scarcity usually results in poor performance of machine learning models on such data. Data augmentation, a common strategy for performance improvement in vision and language tasks, typically underperforms for tabular data due to the lack of explicit symmetries in the input space. To overcome this challenge, we introduce TabMDA, a novel method for manifold data augmentation on tabular data. This method utilises a pre-trained in-context model, such as TabPFN, to map the data into an embedding space. TabMDA performs label-invariant transformations by encoding the data multiple times with varied contexts. This process explores the learned embedding space of the underlying in-context models, thereby enlarging the training dataset. TabMDA is a training-free method, making it applicable to any classifier. We evaluate TabMDA on five standard classifiers and observe significant performance improvements across various tabular datasets. Our results demonstrate that TabMDA provides an effective way to leverage information from pre-trained in-context models to enhance the performance of downstream classifiers. Code is available at `https://github.com/AdrianBZG/TabMDA`.

## 1. Introduction

Tabular data is prevalent and integral to critical domains such as medicine (Balendra and Isaacs, 2018; Kelly and Semsarian, 2009; Meira et al., 2001), physics (Baldi et al., 2014; Kasieczka et al., 2021) and chemistry (Keith et al., 2021; Zhai et al., 2021). However, acquiring large datasets can be prohibitively expensive or impossible, making it challenging to train machine learning models effectively (Jiang et al., 2024; Margeloiu et al., 2023b). In vision and language tasks, a common remedy for addressing model performance issues due to data scarcity is employing data augmentation (DA) techniques to generate additional synthetic samples (Shorten and Khoshgoftaar, 2019). Techniques such as exploiting symmetries in the data, like rotating, flipping or altering the colour of images, or generating paraphrased sentences for text, are used to increase the diversity of the training set, and often lead to improved model generalisation. However, applying DA to tabular data in the input space is challenging, because tabular data is heterogeneous and lacks clear symmetries (Onishi and Meguro, 2023). Consequently, existing tabular augmentation methods in the input space often degrade model performance, hindering their widespread adoption (Manousakas and Aydöre, 2023).

Data augmentation on a learned manifold is advantageous, because it enables label-invariant transformations (Sheng and Xiao, 2022; Wang, 2023). Manifolds provide a structured, continuous space where data points can be meaningfully interpolated or transformed. However, current manifold data augmentation (MDA) methods face two challenges in our context of tabular data. Firstly, these methods are typically model-specific, and no general MDA method is applicable across different models. Secondly, MDA has been primarily developed for neural networks, which learn a manifold space (Fonseca and Bacao, 2023). However, for tabular data, neural networks are often outperformed by more traditional methods such as tree-based algorithms like XGBoost (Chen and Guestrin, 2016) or logistic regression, which operate directly in the input space and do not learn a manifold during training. Therefore, it remains an open question how to enhance these traditional methods with the benefits of MDA.

To address the challenge of small tabular datasets, we introduce TabMDA (Figure 1), a new training-free method for performing manifold data augmentation (MDA) on tabular data. TabMDA is a general method that works with any downstream classifier. It has two components: first, it leverages an existing pre-trained in-context model, such

---

[*]Equal contribution  [1]Department of Computer Science and Technology, University of Cambridge, UK [2]Precision Breast Cancer Institute, Department of Oncology, University of Cambridge, UK. Correspondence to: Andrei Margeloiu <am2770@cam.ac.uk>, Adrián Bazaga <ar989@cam.ac.uk>.

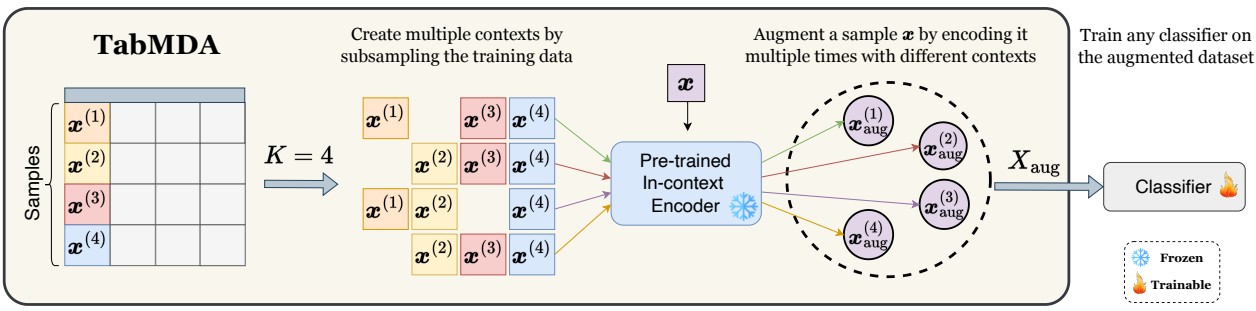

*Figure 1.* TabMDA improves tabular data classifiers by jointly embedding and augmenting the dataset in the embedding space of pre-trained tabular transformers using in-context learning. It generates multiple embeddings for each input by presenting different contexts to the encoder, leveraging its in-context learning capability. This results in an expanded training dataset with more diverse samples, enhancing the accuracy and robustness of the downstream predictor. TabMDA is training-free and can be applied to any classifier.

as TabPFN (Hollmann et al., 2023), to embed the data in a manifold. Second, we introduce in-context subsetting (ICS), a novel technique that uses an in-context classifier to facilitate label-invariant transformations directly in the manifold space. ICS encodes the input data multiple times using an in-context encoder, with each iteration using different data subsets as context for the transformer model. This process explores the manifold space learned by the pre-trained encoder (see Figure 2 for an illustration). The downstream classifier is then trained on this expanded manifold dataset, thereby indirectly incorporating the pre-training knowledge into any subsequent classification model. Crucially, our method requires no additional training and is applicable to any classifier. We investigate TabMDA using five common tabular classifiers across 28 different dataset configurations of varying sizes. Our results show that TabMDA substantially improves the performance of various standard tabular classifiers, including tree-based ensembles such as XGBoost, while reducing the variability among their performance. As a result, TabMDA enables explainable classifiers, such as KNN, to become very competitive and sometimes even achieve the highest performance.

Our contributions can be summarised as:

- **TabMDA:** A novel training-free data augmentation method that jointly embeds and augments the data using pre-trained in-context models. TabMDA can be applied to any classifier, and our results demonstrate that it improves performance across classifiers on small datasets.
- **In-context Subsetting:** A new technique for label-invariant transformations in the manifold of in-context models by using multiple iterations of in-context encoding with different data contexts.

## 2. Related Work

**Using LLMs on tabular data.** The remarkable success of Large Language Models (LLMs), trained on massive text corpora and scaled to unprecedented sizes (Brown et al., 2020; Ouyang et al., 2022), has motivated their adaptation to tabular learning (Wang et al., 2024). Adapting LLMs for tabular data leverages the broad world knowledge already learned, enables in-context learning (ICL) capabilities, and effectively utilises meta-information in tabular data, such as column names (Ye et al., 2024). For instance, LIFT (Dinh et al., 2022) introduced language-interfaced fine-tuning by adapting GPT-3 (Brown et al., 2020) and GPT-J (Wang and Komatsuzaki, 2021) to multiple tabular learning datasets, finding that the performance of fine-tuned LLMs was roughly comparable to traditional solutions. TabLLM (Hegselmann et al., 2023) fine-tuned T0 (Sanh et al., 2022), showing slight underperformance compared to classical tree models. Although these studies demonstrate that while LLMs trained on text corpora can be applied to tabular learning, there remains a performance gap between LLMs and classical tabular models due to the lack of tabular-specific inductive biases in LLMs.

**Transformer-based models for tabular data.** Transformers for tabular data are typically developed by training from scratch on the target task. Occasionally, an additional pre-training phase on the target task itself is included, unlike LLMs, which use large external corpora. These models introduce various attention mechanisms to handle the tabular data structure. For instance, TUTA (Wang et al., 2021) introduces a novel attention mechanism that allows the model to attend to the entire table at once, while TAPAS (Herzig et al., 2020) introduces a column-specific attention mechanism that allows the model to attend to the relevant columns in the table. TransTab (Wang and Sun, 2022) combines column descriptions and cells as input to a gated Transformer model for feature encoding. SAINT (Somepalli et al., 2021) performs attention across both rows and columns to capture the relationships between different cells in the table. More notable examples in this type of models include TabNet (Arık and Pfister, 2021), TabTransformer (Huang et al., 2020), and

Real Input Data      Manifold Space      Manifold Data Augmentation

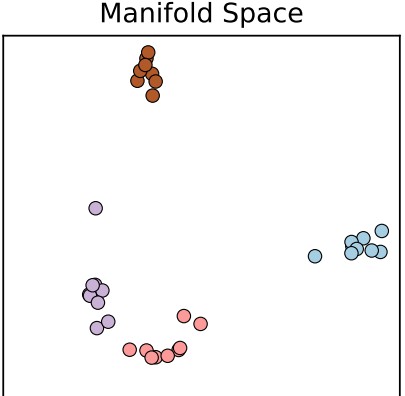

*Figure 2.* PCA linear projection of the first two PCs of the raw input data space for the "vehicle" dataset **(left)**, the manifold space using the encoder from (Hollmann et al., 2023) **(middle)**, and the augmented manifold space after using our proposed method **(right)**. The colour of the data points represents one of four class labels. Visualisations for all datasets are available in Appendix C.

FT-Transformer (Gorishniy et al., 2021). Even though transformers can be competitive on large tabular tasks but are often outperformed by well-tuned MLPs (Kadra et al., 2021), especially on small datasets (Margeloiu et al., 2023a). This suggests that models trained on a single tabular task lack adequate in-context learning (ICL) capabilities for tabular data.

**In-context learning (ICL) for tabular data and TabPFN.** ICL allows Transformer-based models to adapt to new tasks by conditioning on demonstrations of input-output pairs. Recently, ICL has shown promising results on tabular data. Notably, TabPFN (Hollmann et al., 2023), a meta-learned Transformer with ICL capabilities, has demonstrated competitive accuracy compared to tree-based ensembles on small supervised classification tasks. TabPFN employs a Prior-Data Fitted Network (PFN) (Müller et al., 2022) approach, where it is pre-trained on synthetic datasets generated from Structural Causal Models (SCMs) and Bayesian Neural Networks (BNNs), which resembles the causal structure commonly assumed in tabular data. This pretraining enables TabPFN to approximate Bayesian inference in a single forward pass, reducing computational needs and eliminating hyper-parameter tuning. During inference, TabPFN uses the training dataset (context) and new sample features as input, directly outputting the posterior predictive distribution (PPD) for the new samples. This leverages in-context learning to embed new samples without parameter updates, making TabPFN effective on small datasets. Although TabPFN can embed samples into a rich manifold space using its in-context learning abilities, the nature of this embedding space has not been studied. However, we believe these manifolds could facilitate data augmentation for various classifiers.

**Tabular Data Augmentation.** Tabular data augmentation is less explored due to several challenges, such as the lack of explicit symmetries like rotations or translations and gener-

ally weaker feature correlations compared to the strong spatial or semantic relationships in image or text data (Borisov et al., 2022). Augmenting tabular data often involves training generative models, which is computationally expensive (Durkan et al., 2019; Kotelnikov et al., 2023; Liu et al., 2023; Watson et al., 2023; Xu et al., 2019). On small tasks, this can lead to overfitting (Douzas and Bacao, 2018) and mode collapse (Qin et al., 2023; Sampath et al., 2021), where the model fails to capture the diversity of each class. Recently, TabPFGen (Ma et al., 2023) transformed TabPFN into an energy-based class-conditional generator for data augmentation in the input-space. While it does not require specialised training, TabPFGen achieves minimal performance improvements and retains TabPFN's limitations, such as being applicable to only up to ten classes. In contrast, our method, TabMDA, is capable of handling datasets with an unlimited number of classes. Overall, existing methods for augmenting tabular data often yield mixed results and can even degrade model performance (Manousakas and Aydöre, 2023; Seedat et al., 2023).

## 3. Method

We propose a general data augmentation framework TabMDA (Figure 1) that utilises the in-context learning (ICL) capabilities of any pre-trained tabular in-context model to increase the robustness and performance of any downstream machine learning models. In a nutshell, our approach augments the training dataset by introducing diversity through label-invariant transformations in the manifold space, leveraging multiple contexts for the embedder. We then train the downstream prediction model using this augmented dataset.

**Problem Setup:** We focus on supervised classification tasks involving $\mathcal{Y}$ classes. Consider a $D$-dimensional

dataset $\{(\boldsymbol{x}^{(i)}, y_i)\}$, where each sample $\boldsymbol{x}^{(i)} \in \mathbb{R}^D$ is continuous, and its corresponding label $y_i \in \mathcal{Y}$ is categorical. The training data can be represented as a matrix $\boldsymbol{X}_{\text{train}} := [\boldsymbol{x}^{(1)}, \boldsymbol{x}^{(2)}, ..., \boldsymbol{x}^{(N)}]^\top \in \mathbb{R}^{N \times D}$, where each row corresponds to a sample, and a label vector $\boldsymbol{y}_{\text{train}} := [y_1, y_2, ..., y_N]$ containing the labels of the training samples.

**Embedding the data using In-Context Encoders.** TabMDA begins by embedding the real data into a manifold space of a pre-trained tabular in-context model. In this work, we specifically apply TabMDA to TabPFN (Hollmann et al., 2023), a pre-trained tabular transformer. TabPFN is designed to approximate the posterior predictive distribution (PPD) with a tabular-specific prior in a single forward pass. This pretraining is conducted on synthetic data generated from Structural Causal Models (SCMs) and Bayesian Neural Networks (BNNs), which mimic the causal structures typically found in tabular data. In particular, we utilise only the encoder part of the TabPFN model for embedding data points $\boldsymbol{x}$ in its latent space of size $D'$. The TabPFN's encoder takes the input samples $\boldsymbol{X}_{\text{train}}$ and their labels $y_{\text{train}}$ as the context. For clarity, we denote the context $\boldsymbol{X}_{\text{ctx}}$ using only the input samples $\boldsymbol{X}_{\text{train}}$. TabPFN, a transformer-based model, encodes data points $\boldsymbol{x}$ by applying attention between the data point and its context $\boldsymbol{X}_{\text{ctx}}$. This process implies that the final embedding $\phi(\boldsymbol{X}_{\text{ctx}}, \boldsymbol{x})$ is context-dependent. Consequently, fitting TabPFN with different datasets (contexts) will produce different embeddings for the same data point $\boldsymbol{x}$.

**In-context Subsetting (ICS).** We introduce a manifold data augmentation which performs label-invariant transformations within the manifold (Algorithm 1). The core idea of ICS leverages the context-dependent nature of embeddings. By slightly varying the contexts, ICS generates variations in the embeddings (Figure 2). Specifically, for each data point $\boldsymbol{x}$ to be augmented, we generate $K$ different contexts by stratified sampling $N_{\text{ctx}}$ points from the training data $\boldsymbol{X}_{\text{train}}$ without replacement. Each context is denoted as $\boldsymbol{X}_{\text{ctx}}^{(k)} := \{\boldsymbol{x}^{(i_k)}\}_{i_k=1}^{N_{\text{ctx}}}$ for $k \in \{1, \ldots, K\}$. TabMDA encodes the data point $\boldsymbol{x}$ with each context $\boldsymbol{X}_{\text{ctx}}^{(k)}$, producing an augmented embedding $\boldsymbol{x}_{\text{aug}}^{(k)}$. This use of random contexts for encoding enhances data diversity in the manifold space. ICS produces $K$ augmented samples $\boldsymbol{X}_{\text{aug}} := [\boldsymbol{x}_{\text{aug}}^{(1)}, \boldsymbol{x}_{\text{aug}}^{(2)}, \ldots, \boldsymbol{x}_{\text{aug}}^{(K)}]$ for each input.

**Training any Downstream Classifier** with TabMDA starts by embedding and augmenting the training set using ICS. Specifically, we encode each training point $\boldsymbol{x}^{(i)}$ to obtain the augmented training dataset:

$$\boldsymbol{X}_{\text{train}}^{\text{augm}} := \left\{ \left( \text{ICS}\left(\phi, K, N_{\text{ctx}}, \boldsymbol{X}_{\text{train}}, \boldsymbol{x}^{(i)}\right), \underbrace{y_i, ..., y_i}_{K \text{ times}} \right) \right\}_{i=1}^{N}$$

where $\boldsymbol{X}_{\text{train}}^{\text{augm}} \in \mathbb{R}^{(K \times N) \times D'}$ and $D'$ is the dimensionality

---

**Algorithm 1** In-context Subsetting (ICS)

**Input:** Tabular in-context model $\phi$; Number of contexts to generate $K$; Context size $N_{\text{ctx}}$; Train data $\boldsymbol{X}_{\text{train}}$; Data point to augment $\boldsymbol{x}$
**Output:** Augmented versions of $\boldsymbol{x}$
1: **function** ICS($\phi, K, N_{\text{ctx}}, \boldsymbol{X}_{\text{train}}, \boldsymbol{x}$)
2:     **for** $k = 1$ to $K$ **do**
3:         $\boldsymbol{X}_{\text{ctx}}^{(k)}$ = Stratified sample $N_{\text{ctx}}$ points from $\boldsymbol{X}_{\text{train}}$
4:         $\boldsymbol{x}_{\text{aug}}^{(k)} = \phi(\boldsymbol{X}_{\text{ctx}}^{(k)}, \boldsymbol{x})$     ▷ Augment the data point
5:     **end for**
6:     Let $\boldsymbol{X}_{\text{aug}} := [\boldsymbol{x}_{\text{aug}}^{(1)}, \boldsymbol{x}_{\text{aug}}^{(2)}, \ldots, \boldsymbol{x}_{\text{aug}}^{(K)}]$
7:     **return** $\boldsymbol{X}_{\text{aug}}$ as the augmented versions of $\boldsymbol{x}$
8: **end function**

---

of the encoder's latent space. We assume ICS is label-invariant, so all ICS embedding retains the same label as the original sample. The downstream classifier is trained exclusively on the augmented dataset $\boldsymbol{X}_{\text{train}}^{\text{augm}}$, which is the *embedding space* and is $K$ times larger than the initial dataset $\boldsymbol{X}_{\text{train}}$. This augmented dataset is more diverse and includes information from the encoding manifold learned during the pre-training of the in-context encoder. The test data points $\boldsymbol{x}^{\text{test}}$ are encoded using the full training data, which is equivalent to encoding using ICS $(\phi, 1, N, \boldsymbol{X}_{\text{train}}, \boldsymbol{x}^{\text{test}})$. As we next demonstrate, TabMDA with ICS improves generalisation and reduces performance variability across classifiers.

# 4. Experiments

## 4.1. Experimental Setup

**Datasets.** We focus on small-to-medium classification tasks, because this regime allows the study of the effect of data augmentation. We consider six tabular datasets, with complete details included in Appendix A.1. We evaluate all models over 10 runs and report the mean and standard deviation of the test-balanced accuracy averaged across them.

**Setup and evaluation.** For each dataset comprising $N$ samples, we initially split it into stratified training and test sets. We use a sizeable test set to ensure an accurate evaluation of the model's performance. The size of the test set is calculated as $N_{\text{test}} = \min\left(\frac{N}{2}, 500\right)$. To better understand the impact of our method across different dataset sizes, we subsample the full training set to simulate varying levels of data availability, creating subsets with sample sizes $N_{\text{real}} \in \{20, 50, 100, 200, 500\}$. Each subset is split into training and validation sets, with 80% of $N_{\text{real}}$ for training and the remaining 20% for validation. We repeat the training set splitting 10 times to result in 10 runs per subset size. Note that the same test set is used for all repeats to ensure consistency in performance evaluation.

**TabMDA settings.** For the in-context model, we use the encoder component of TabPFN (Hollmann et al., 2023), a pre-trained classifier. We utilise the embeddings produced by its transformer encoder and we use the official weights

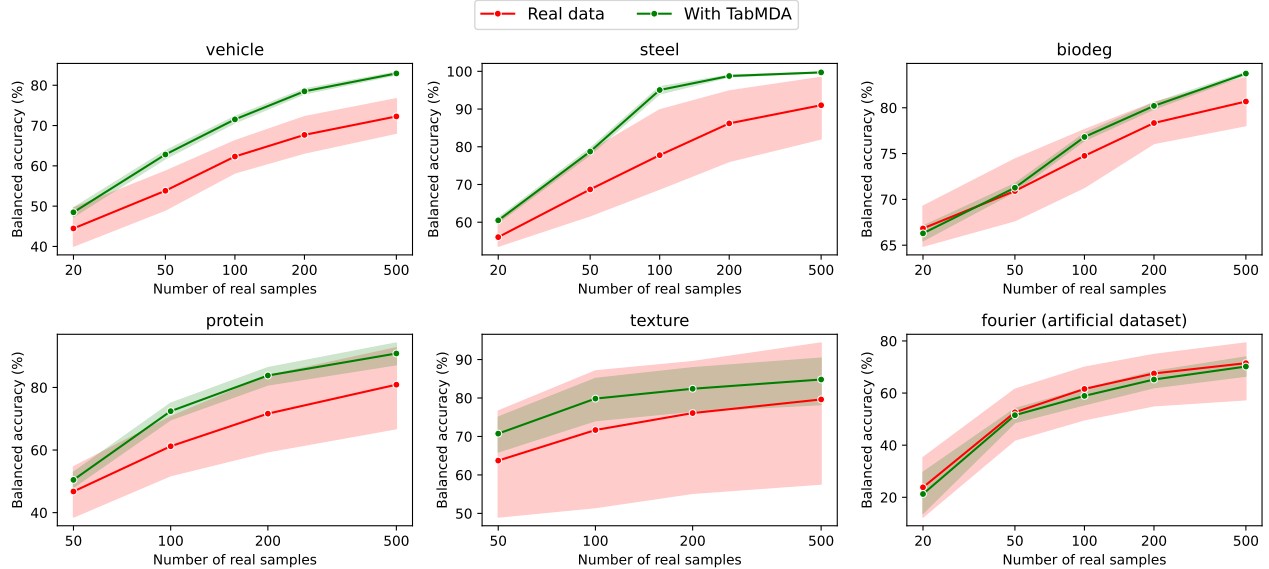

*Figure 3.* Average accuracy (%) for five downstream classifiers trained on real data (the original input space) and TabMDA embeddings. We report the mean±std of test balanced accuracy over 10 runs for each predictor, totalling 50 runs. Training with TabMDA substantially improves performance across five real-world tabular datasets and reduces variability among classifiers. It also performs well on artificial datasets, such as "fourier", which are processed images, despite differing from the inductive bias of the in-context encoder, TabPFN.

released by the TabPFN authors. We use TabPFN without ensembling (referred to as TabPFN$_{n.e.}$ in the original paper), and we do not disable its internal data preprocessing. For the proposed in-context subsetting, we define the context size as a proportion of the training set, considering context sizes of 0.5, 0.7, 0.9, and 1. Additionally, we evaluate 5, 20, and 50 subcontexts. We select the optimal TabMDA settings based on the validation balanced accuracy for each dataset configuration.

### 4.2. Impact on the Classification Performance

We assess the impact of TabMDA on the performance of various tabular classifiers. We train these classifiers on real data (the original input space) and on TabMDA-augmented data, which consists of augmented embeddings output from the TabPFN's transformer encoder. We consider five classification models with different inductive biases: distance-based methods (K-Nearest Neighbors (KNN)), linear methods (Logistic Regression) and tree-based methods (Decision Tree (Breiman et al., 1984), Random Forest (Breiman, 2001) and XGBoost (Chen and Guestrin, 2016)). We could not use TabPFN for the final prediction because it handles data with up to 100 dimensions, and the embeddings provided by TabMDA have 512 dimensions. Following previous work (Van Breugel et al., 2023), we train the predictors with a fixed set of hyper-parameters, which we provide in Appendix A.2. We use identical hyper-parameters for the downstream models when trained on real data and with TabMDA.

**On the overall downstream performance.** Figure 3 illustrates the overall impact of training downstream predictors with TabMDA. The results indicate that TabMDA enhances overall performance by up to 15% across five real-world tabular datasets. Improvements are observed across all dataset sizes, with more significant gains for datasets of 100 samples or more. On average, 3% of this performance improvement is attributed to the proposed in-context subsetting (regardless of the context sizes), and the remaining performance gain is due to training directly on the TabPFN's manifold (see Appendix B). Higher context sizes of 0.7 and 0.9 tend to yield better results than the smaller context size of 0.5. This trend is particularly noticeable for small datasets, where a smaller context size might introduce too much variance, negatively impacting performance (see Appendix B).

One important observation is that training with TabMDA significantly reduces performance variability across classifiers. For example, on the "vehicle", "steel" and "biodeg" datasets, classifiers trained on real data exhibit considerable performance variation, as indicated by the large standard deviations in accuracy (marked with red hue band in Figure 3). In contrast, training with TabMDA substantially reduces the performance variability, resulting in predictors with similar performance levels. This is advantageous as it enables using interpretable classifiers, such as KNN, which typically perform poorly when applied directly to the input space.

As expected, the performance of TabMDA depends on the underlying in-context model and its pre-training inductive

*Table 1.* Classification accuracy (%) of five downstream predictors trained on real data (the original input space) and on TabMDA embeddings. We present the mean ± std of the test balanced accuracy averaged over ten runs. ↑ indicates that TabMDA improves the performance of the downstream classifier. N/A indicates that TabPFN is not applicable to "texture". We **bold** the highest accuracy for each dataset of different sample sizes. Our method, TabMDA, consistently improves the accuracy of downstream classifiers, with an average improvement of up to 10%.

| Dataset | $N_{\text{real}}$ | TabPFN | KNN | | Logistic Regression | | Decision Tree | | Random Forest | | XGBoost | |
|---|---|---|---|---|---|---|---|---|---|---|---|---|
| | | | Real data | TabMDA | Real data | TabMDA | Real data | TabMDA | Real data | TabMDA | Real data | TabMDA |
| vehicle | 20 | $50.81_{\pm6.38}$ | $38.68_{\pm3.59}$ | $51.12_{\pm4.44}$ ↑ | $52.40_{\pm3.64}$ | $51.69_{\pm3.44}$ | $38.87_{\pm5.91}$ | $41.44_{\pm6.76}$ ↑ | $47.16_{\pm3.95}$ | $\mathbf{52.84_{\pm5.28}}$ ↑ | $45.21_{\pm4.60}$ | $51.54_{\pm5.44}$ ↑ |
| | 50 | $64.55_{\pm5.54}$ | $49.32_{\pm3.51}$ | $64.93_{\pm3.66}$ ↑ | $60.99_{\pm3.87}$ | $\mathbf{65.62_{\pm3.52}}$ ↑ | $46.42_{\pm5.40}$ | $59.89_{\pm4.93}$ ↑ | $55.76_{\pm3.60}$ | $64.21_{\pm3.55}$ ↑ | $56.60_{\pm4.03}$ | $63.64_{\pm4.73}$ ↑ |
| | 100 | $\mathbf{73.67_{\pm2.93}}$ | $59.04_{\pm3.86}$ | $73.31_{\pm1.94}$ ↑ | $68.80_{\pm2.80}$ | $72.10_{\pm2.97}$ ↑ | $55.52_{\pm6.49}$ | $69.76_{\pm1.88}$ ↑ | $62.56_{\pm5.54}$ | $73.48_{\pm3.20}$ ↑ | $65.65_{\pm3.67}$ | $71.66_{\pm2.24}$ ↑ |
| | 200 | $81.13_{\pm1.87}$ | $64.31_{\pm2.78}$ | $77.74_{\pm1.75}$ ↑ | $74.26_{\pm1.82}$ | $80.77_{\pm1.95}$ ↑ | $65.75_{\pm2.85}$ | $81.14_{\pm1.16}$ ↑ | $66.96_{\pm3.57}$ | $\mathbf{81.58_{\pm2.20}}$ ↑ | $72.74_{\pm2.00}$ | $80.50_{\pm2.56}$ ↑ |
| | 500 | $84.11_{\pm1.00}$ | $68.90_{\pm0.92}$ | $83.33_{\pm1.02}$ ↑ | $78.29_{\pm1.11}$ | $84.57_{\pm0.85}$ ↑ | $65.75_{\pm2.85}$ | $81.14_{\pm1.16}$ ↑ | $70.07_{\pm0.94}$ | $\mathbf{85.11_{\pm1.32}}$ ↑ | $78.30_{\pm1.67}$ | $84.12_{\pm1.21}$ ↑ |
| steel | 20 | $57.96_{\pm4.29}$ | $54.17_{\pm3.72}$ | $61.62_{\pm3.62}$ ↑ | $63.96_{\pm9.21}$ | $\mathbf{65.10_{\pm4.00}}$ ↑ | $53.92_{\pm3.12}$ | $61.08_{\pm6.68}$ ↑ | $53.23_{\pm2.73}$ | $64.76_{\pm5.27}$ ↑ | $54.94_{\pm3.48}$ | $62.03_{\pm4.41}$ ↑ |
| | 50 | $82.09_{\pm7.91}$ | $69.81_{\pm5.54}$ | $81.75_{\pm7.27}$ ↑ | $\mathbf{88.06_{\pm5.64}}$ | $82.42_{\pm9.58}$ | $62.68_{\pm9.97}$ | $76.10_{\pm7.92}$ ↑ | $60.40_{\pm3.43}$ | $82.69_{\pm8.38}$ ↑ | $62.53_{\pm2.57}$ | $81.55_{\pm7.17}$ ↑ |
| | 100 | $97.37_{\pm1.37}$ | $81.55_{\pm4.73}$ | $97.76_{\pm1.28}$ ↑ | $\mathbf{98.85_{\pm1.20}}$ | $97.78_{\pm1.46}$ | $70.78_{\pm12.39}$ | $95.25_{\pm2.98}$ ↑ | $63.89_{\pm2.75}$ | $98.04_{\pm1.29}$ ↑ | $73.79_{\pm6.19}$ | $96.99_{\pm1.27}$ ↑ |
| | 200 | $98.84_{\pm0.70}$ | $87.58_{\pm2.98}$ | $\mathbf{99.57_{\pm0.62}}$ ↑ | $99.43_{\pm0.58}$ | $99.55_{\pm0.61}$ ↑ | $84.40_{\pm8.69}$ | $97.83_{\pm1.86}$ ↑ | $68.29_{\pm2.91}$ | $99.31_{\pm0.69}$ ↑ | $91.27_{\pm3.47}$ | $98.79_{\pm1.01}$ ↑ |
| | 500 | $99.77_{\pm0.30}$ | $93.51_{\pm1.22}$ | $\mathbf{99.91_{\pm0.14}}$ ↑ | $99.78_{\pm0.27}$ | $99.83_{\pm0.24}$ ↑ | $88.01_{\pm0.71}$ | $99.77_{\pm0.23}$ ↑ | $74.22_{\pm2.02}$ | $99.83_{\pm0.24}$ ↑ | $99.55_{\pm0.72}$ | $99.67_{\pm0.50}$ ↑ |
| biodeg | 20 | $66.85_{\pm9.33}$ | $67.24_{\pm6.19}$ | $69.64_{\pm5.19}$ ↑ | $71.63_{\pm5.43}$ | $69.04_{\pm6.99}$ | $64.15_{\pm7.02}$ | $63.48_{\pm7.03}$ | $65.03_{\pm7.75}$ | $\mathbf{72.06_{\pm4.04}}$ ↑ | $66.10_{\pm6.95}$ | $69.82_{\pm4.61}$ ↑ |
| | 50 | $75.21_{\pm2.67}$ | $73.27_{\pm2.31}$ | $72.37_{\pm3.37}$ | $\mathbf{76.42_{\pm3.03}}$ | $74.83_{\pm3.24}$ | $64.85_{\pm3.55}$ | $70.01_{\pm6.57}$ ↑ | $69.47_{\pm3.52}$ | $73.28_{\pm6.31}$ ↑ | $70.56_{\pm4.05}$ | $72.90_{\pm6.44}$ ↑ |
| | 100 | $78.78_{\pm1.56}$ | $76.98_{\pm1.77}$ | $78.59_{\pm1.92}$ ↑ | $\mathbf{79.06_{\pm1.66}}$ | $77.37_{\pm2.61}$ | $68.86_{\pm3.61}$ | $77.08_{\pm3.57}$ ↑ | $73.88_{\pm2.36}$ | $77.85_{\pm2.77}$ ↑ | $74.98_{\pm2.08}$ | $78.02_{\pm2.84}$ ↑ |
| | 200 | $\mathbf{82.66_{\pm1.87}}$ | $79.86_{\pm2.50}$ | $82.64_{\pm1.80}$ ↑ | $81.92_{\pm1.56}$ | $80.68_{\pm2.44}$ | $75.12_{\pm3.24}$ | $80.17_{\pm4.61}$ ↑ | $75.48_{\pm2.09}$ | $81.34_{\pm1.04}$ ↑ | $79.30_{\pm1.41}$ | $80.13_{\pm1.84}$ ↑ |
| | 500 | $84.96_{\pm0.67}$ | $82.48_{\pm0.65}$ | $\mathbf{85.05_{\pm0.81}}$ ↑ | $83.86_{\pm0.54}$ | $84.47_{\pm1.24}$ ↑ | $77.12_{\pm1.89}$ | $82.54_{\pm1.13}$ ↑ | $76.78_{\pm2.01}$ | $84.45_{\pm0.68}$ ↑ | $83.17_{\pm1.63}$ | $84.09_{\pm1.01}$ ↑ |
| protein | 50 | $51.65_{\pm5.74}$ | $36.68_{\pm3.40}$ | $55.82_{\pm4.60}$ ↑ | $61.22_{\pm3.56}$ | $\mathbf{62.48_{\pm4.21}}$ ↑ | $38.12_{\pm2.82}$ | $40.51_{\pm5.98}$ ↑ | $52.98_{\pm3.13}$ | $54.90_{\pm4.22}$ ↑ | $44.78_{\pm4.01}$ | $56.20_{\pm5.22}$ ↑ |
| | 100 | $72.58_{\pm3.76}$ | $50.28_{\pm3.67}$ | $79.26_{\pm2.66}$ ↑ | $79.46_{\pm3.22}$ | $\mathbf{80.44_{\pm2.67}}$ ↑ | $48.11_{\pm5.03}$ | $57.89_{\pm3.25}$ ↑ | $63.69_{\pm2.74}$ | $77.32_{\pm2.96}$ ↑ | $74.98_{\pm1.44}$ | $91.49_{\pm1.34}$ ↑ |
| | 200 | $86.69_{\pm2.77}$ | $66.42_{\pm2.62}$ | $91.05_{\pm1.81}$ ↑ | $90.98_{\pm1.79}$ | $\mathbf{92.97_{\pm1.91}}$ ↑ | $50.50_{\pm2.26}$ | $65.40_{\pm4.73}$ ↑ | $70.89_{\pm3.18}$ | $88.40_{\pm1.41}$ ↑ | $79.38_{\pm1.44}$ | $91.49_{\pm1.34}$ ↑ |
| | 500 | $95.58_{\pm0.98}$ | $84.94_{\pm1.58}$ | $98.09_{\pm0.53}$ ↑ | $97.85_{\pm0.81}$ | $\mathbf{98.21_{\pm0.59}}$ ↑ | $53.48_{\pm3.19}$ | $70.43_{\pm1.54}$ ↑ | $75.90_{\pm2.09}$ | $97.27_{\pm0.93}$ ↑ | $92.29_{\pm1.42}$ | $98.06_{\pm0.60}$ ↑ |
| texture | 50 | N/A | $62.81_{\pm3.09}$ | $78.50_{\pm3.85}$ ↑ | $86.07_{\pm2.67}$ | $\mathbf{88.08_{\pm2.53}}$ ↑ | $37.34_{\pm7.19}$ | $44.49_{\pm4.58}$ ↑ | $70.41_{\pm2.01}$ | $79.24_{\pm3.31}$ ↑ | $62.03_{\pm4.78}$ | $83.48_{\pm3.25}$ ↑ |
| | 100 | N/A | $77.84_{\pm2.29}$ | $93.75_{\pm1.48}$ ↑ | $94.05_{\pm1.75}$ | $\mathbf{95.52_{\pm1.18}}$ ↑ | $34.84_{\pm3.71}$ | $35.55_{\pm5.36}$ ↑ | $76.42_{\pm1.46}$ | $90.67_{\pm1.90}$ ↑ | $75.22_{\pm3.83}$ | $94.04_{\pm1.37}$ ↑ |
| | 200 | N/A | $85.09_{\pm1.69}$ | $96.55_{\pm1.18}$ ↑ | $95.62_{\pm1.40}$ | $\mathbf{97.49_{\pm0.85}}$ ↑ | $38.12_{\pm4.61}$ | $39.73_{\pm3.88}$ ↑ | $76.65_{\pm1.34}$ | $94.25_{\pm1.22}$ ↑ | $84.00_{\pm1.83}$ | $96.39_{\pm1.19}$ ↑ |
| | 500 | N/A | $91.28_{\pm1.32}$ | $98.11_{\pm0.25}$ ↑ | $98.01_{\pm0.35}$ | $\mathbf{98.73_{\pm0.47}}$ ↑ | $39.67_{\pm3.72}$ | $41.46_{\pm7.53}$ ↑ | $77.34_{\pm2.20}$ | $97.40_{\pm0.63}$ ↑ | $91.83_{\pm1.29}$ | $98.45_{\pm0.51}$ ↑ |
| fourier (artificial) | 20 | $26.58_{\pm5.16}$ | $19.58_{\pm2.17}$ | $18.32_{\pm3.04}$ | $\mathbf{42.88_{\pm5.23}}$ | $35.00_{\pm5.65}$ | $11.26_{\pm2.78}$ | $13.72_{\pm2.67}$ ↑ | $35.42_{\pm3.19}$ | $29.29_{\pm3.98}$ | $10.00_{\pm0.00}$ | $10.00_{\pm0.00}$ |
| | 50 | $55.38_{\pm4.83}$ | $54.38_{\pm2.56}$ | $57.16_{\pm3.99}$ ↑ | $60.66_{\pm1.64}$ | $63.38_{\pm2.51}$ ↑ | $31.64_{\pm4.53}$ | $36.64_{\pm3.62}$ ↑ | $\mathbf{65.78_{\pm3.55}}$ | $55.16_{\pm4.32}$ | $50.62_{\pm5.22}$ | $53.72_{\pm3.01}$ ↑ |
| | 100 | $61.98_{\pm3.40}$ | $63.24_{\pm2.17}$ | $68.54_{\pm2.45}$ ↑ | $67.84_{\pm2.50}$ | $70.10_{\pm3.07}$ ↑ | $38.68_{\pm6.94}$ | $40.28_{\pm4.41}$ ↑ | $\mathbf{73.20_{\pm2.61}}$ | $60.48_{\pm3.24}$ | $64.96_{\pm3.16}$ | $67.86_{\pm3.16}$ ↑ |
| | 200 | $68.94_{\pm2.41}$ | $70.74_{\pm2.57}$ | $75.10_{\pm2.06}$ ↑ | $72.84_{\pm2.43}$ | $74.62_{\pm2.36}$ ↑ | $43.62_{\pm5.61}$ | $42.94_{\pm5.87}$ | $\mathbf{76.42_{\pm2.53}}$ | $68.24_{\pm1.14}$ | $74.00_{\pm1.56}$ | $74.74_{\pm1.67}$ ↑ |
| | 500 | $76.28_{\pm2.07}$ | $78.08_{\pm1.04}$ | $81.22_{\pm0.93}$ ↑ | $77.52_{\pm1.17}$ | $79.88_{\pm1.76}$ ↑ | $43.82_{\pm6.07}$ | $41.98_{\pm6.59}$ | $77.64_{\pm1.51}$ | $76.42_{\pm1.86}$ | $80.30_{\pm1.51}$ | $\mathbf{81.24_{\pm0.76}}$ ↑ |
| **Average accuracy** | | 73.77 | 67.43 | 77.50↑ | 78.70 | 79.38↑ | 53.06 | 60.83↑ | 67.00 | 77.14↑ | 69.59 | 77.16↑ |

bias (i.e., the model's prior). We use TabPFN, which was trained on data with a suitable prior for tabular data, allowing it to perform well with real-world datasets. Note that, from the six selected datasets in Figure 3, "fourier" is an artificial dataset consisting of Fourier coefficients from digitised images of handwritten numerals on Dutch maps. This dataset does not align well with TabPFN's prior, leading to minimal performance improvements by TabMDA. Despite this, TabMDA significantly enhances performance stability.

**TabMDA improves performance across classifiers.** The results in Table 1 show substantial improvements in classification accuracy across multiple datasets when using TabMDA embeddings compared to training on real data alone. TabMDA consistently enhances performance for common tree-based ensembles like Random Forest and XGBoost. Logistic Regression also sees improvements, albeit to a lesser extent. Notably, training with TabMDA achieves the highest performance in 18 out of 24 real-world dataset settings (as "fourier" is an artificial dataset).

**KNN trained with TabMDA is surprisingly competitive.** Simple classifiers like KNN are advantageous due to their explainability and ease of debugging. KNN makes predictions based on available training data, making them easy to understand and debug. These classifiers offer highly desir-

able properties, such as providing prototypical explanations of predictions. However, when trained on real data, KNN shows a notable performance gap of ≈9% when compared to other classifiers. The results in Table 1 show that KNN trained with TabMDA is surprisingly competitive, closely matching maximal performance. On a larger dataset of 500 samples, KNN (with TabMDA) achieves the best accuracy on two datasets and encounters less than 2% performance degradation on the other three. This result indicates that the manifolds learned by the pre-trained in-context transforms possess meaningful distances, making it an interesting direction for future work.

**Limitations and Future Work.** TabMDA inherits several limitations from its underlying in-context model encoder. In this study, we utilised TabPFN, which can handle tabular data with up to 100 features. As advancements in these tabular in-context models occur, we anticipate corresponding improvements in TabMDA. A significant limitation is that the in-context model requires access to the entire training set for inference, which can be impractical or impossible at test time due to privacy concerns. To address this, one potential solution is knowledge distillation of the underlying encoder, which has been shown to retain most of its performance (Müller et al., 2023). Future research could explore

the embedding spaces of tabular in-context models and the effects of the proposed in-context subsetting.

## 5. Conclusions

In this paper, we introduced TabMDA, a novel training-free manifold data augmentation method for tabular data that leverages a pre-trained in-context model, such as TabPFN, and incorporates our in-context subsetting (ICS) for label-invariant transformations. TabMDA significantly improves the performance of standard tabular classifiers by up to 15% and allows simpler classifiers like KNN to achieve competitive results with minimal performance loss. Notably, training with TabMDA leads to more consistent and reliable results by significantly reducing the performance variability across classifiers. The improvements in performance and stability enable explainable classifiers like KNN to become very competitive and sometimes even achieve the highest performance. TabMDA is applicable to any downstream classifier and does not require dedicated training.

## Acknowledgements

The authors would like to thank Xiangjian Jiang for making the dataset splits. AM acknowledges the support from the Cambridge ESRC Doctoral Training Partnership. NS acknowledges the support of the U.S. Army Medical Research and Development Command of the Department of Defense; through the FY22 Breast Cancer Research Program of the Congressionally Directed Medical Research Programs, Clinical Research Extension Award GRANT13769713. Opinions, interpretations, conclusions, and recommendations are those of the authors and are not necessarily endorsed by the Department of Defense.

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

# - Supplementary material -
## TabMDA: Tabular Manifold Data Augmentation for Any Classifier using Transformers with In-context Subsetting

## A. Reproducibility

### A.1. Datasets

All datasets are publicly available on OpenML [Bischl et al., 2021], and their details are listed in Table 2.

*Table 2.* Details of the datasets used for experiments. "fourier" is an artificial dataset created by processing images.

| Dataset | OpenML ID | # samples (N) | # features (D) | N/D | # classes | # min samples per class | # max samples per class |
|---------|-----------|---------------|----------------|-----|-----------|-------------------------|-------------------------|
| vehicle | 54 | 846 | 18 | 47 | 4 | 199 | 218 |
| steel | 1504 | 1941 | 33 | 59 | 2 | 673 | 1268 |
| biodeg | 1494 | 1055 | 41 | 26 | 2 | 356 | 699 |
| protein | 40966 | 1080 | 77 | 14 | 8 | 105 | 150 |
| texture | 40499 | 5500 | 40 | 138 | 11 | 500 | 500 |
| fourier | 14 | 2000 | 76 | 26 | 10 | 200 | 200 |

### A.2. Implementation, Software and Computing Resources

**Software implementation.** Using PyTorch 1.13 [Paszke et al., 2019], an open-source deep learning library with a BSD licence. All numerical plots and graphics have been generated using Matplotlib 3.8 [Hunter, 2007], a Python-based plotting library with a BSD licence. The model architecture was generated using draw.io, `https://github.com/jgraph/drawio`, a free drawing software under Apache License 2.0.

**Implementation of classifier models.** For implementing the various classifier models in our experiments, we utilized established libraries to ensure reliability and consistency. The K-Nearest Neighbors (KNN), Logistic Regression, Decision Tree, and Random Forest classifiers were all implemented using the scikit-learn library [Pedregosa et al., 2011] (BSD license), version 1.3.2. For the XGBoost classifier, we used the xgboost library, version 1.7.6, which is specifically optimized for gradient boosting.

**Hyper-parameters for benchmark models.** The K-Nearest Neighbors (KNN) classifier was configured with 5 neighbors and used the cosine distance metric for measuring similarity between instances. For Logistic Regression, we set the maximum number of iterations to 1000 to allow sufficient convergence during optimization. The Decision Tree classifier was limited to a maximum depth of 3, with a minimum of 2 samples required per leaf to prevent overfitting. Similarly, the Random Forest classifier comprised 200 trees, each with a maximum depth of 3 and a minimum of 2 samples per leaf, balancing complexity and generalization. Lastly, the XGBoost classifier was configured with 200 estimators, a learning rate of 0.3, and a maximum depth of 3.

**Computing Resources.** All our experiments are run on a single machine from an internal cluster with a GPU Nvidia Quadro RTX 8000 with 48GB memory and an Intel(R) Xeon(R) Gold 5218 CPU with 16 cores (at 2.30GHz). The operating system was Ubuntu 20.4.4 LTS.

## B. Impact of In-context Subsetting

Table 3 shows the impact of in-context subsetting (ICS) on classification performance. We compared the performance of using the full context (i.e., no ICS) against ICS with various context sizes $N_{ctx}$. Even though the optimal context subset size varies across datasets, the results show ICS improves performance across datasets compared to using the full context. Higher context sizes of 0.7 and 0.9 tend to yield better results than the smaller context size of 0.5. This trend is particularly noticeable for small datasets, where a smaller context size might introduce too much variance, negatively impacting performance. Therefore, it is advisable to consider larger context sizes of 0.7 and 0.9 for improved classification accuracy, especially when dealing with limited data.

We recognize that one limitation of TabMDA is the need to tune the context size hyper-parameter. Inspired by TrivialAugment [Müller and Hutter, 2021], we implemented a similar augmentation strategy within TabMDA. Instead of using a fixed context size, we randomly sample $N_{ctx} \sim U[0.5, 0.99]$ for each context subsetting. This approach (Column TabMDA (TA) in Table 3) consistently outperforms using the full context, but performs slightly worse than using a fixed context size value. Further investigation is necessary to potentially develop TabMDA into a hyper-parameter-free method.

*Table 3.* Ablation of the context size hyper-parameter for In-context Subsetting (ICS). The last column depicts a TrivialAugment-inspired version of TabMDA, where the context size is uniformly sampled from $N_{ctx} \sim U[0.5, 0.99]$ instead of fixed to a predefined value. ICS improves performance across datasets compared to using the full context.

| Dataset | $N_{real}$ | Full context (no ICS) | TabMDA (with ICS) | | | |
|---|---|---|---|---|---|---|
| | | | $N_{ctx} = 0.5$ | $N_{ctx} = 0.7$ | $N_{ctx} = 0.9$ | $N_{ctx} \sim U[0.5, 0.99]$ |
| vehicle | 20 | 46.47 | 48.56 | 49.97 | **50.43** | 46.28 |
| | 50 | 60.03 | 63.39 | **64.22** | 63.52 | 61.61 |
| | 100 | 68.39 | 72.05 | **72.71** | 72.29 | 70.31 |
| | 200 | 77.14 | 78.7 | **79.04** | 78.85 | 78.03 |
| | 500 | 83.04 | **83.07** | **83.07** | 82.97 | 82.54 |
| steel | 20 | 60.04 | 59.62 | 59.4 | **62.63** | 60.73 |
| | 50 | 77.57 | 76.23 | 79.92 | **80.9** | 78.31 |
| | 100 | 91.57 | 94.9 | 96.6 | **96.85** | 94.32 |
| | 200 | 97.78 | 97.87 | **99.22** | 99.21 | 98.94 |
| | 500 | 99.27 | 99.85 | **99.87** | 99.73 | 99.68 |
| biodeg | 20 | 66.25 | 65.94 | 64.32 | **69.55** | 66.93 |
| | 50 | **72.16** | 69.74 | 71.53 | 71.92 | 71.15 |
| | 100 | 77.03 | 76.47 | 77.13 | **77.33** | 75.5 |
| | 200 | **81.23** | 80.68 | 80.86 | 80.28 | 79.05 |
| | 500 | **84.15** | 83.91 | 83.81 | 83.67 | 83.37 |
| protein | 50 | 45.55 | 50.96 | 53.2 | **53.97** | 48.83 |
| | 100 | 64.34 | 74.65 | **74.75** | 73.44 | 71.78 |
| | 200 | 77.9 | 85.24 | **85.73** | 82.97 | 83.99 |
| | 500 | 86.55 | 92.3 | **92.41** | 88.89 | 91.91 |
| texture | 50 | 66.0 | **73.28** | 72.1 | 72.1 | 69.04 |
| | 100 | 75.53 | 80.82 | **81.07** | 80.41 | 79.88 |
| | 200 | 80.66 | 82.55 | **83.37** | 82.93 | 82.03 |
| | 500 | **85.78** | 84.64 | 84.77 | 85.06 | 84.59 |
| fourier (artificial) | 50 | 44.02 | 51.04 | **53.72** | 53.69 | 50.94 |
| | 100 | 52.6 | **60.76** | 60.51 | 59.47 | 58.28 |
| | 200 | 61.05 | **66.59** | 66.44 | 65.35 | 64.73 |
| | 500 | 68.82 | **70.63** | 70.35 | 70.18 | 69.87 |
| **Average accuracy** | | 72.26 | 74.98 | 75.56 | 75.50 | 74.17 |

## C. PCA plots of real input data, manifold space and manifold data augmentation using TabMDA

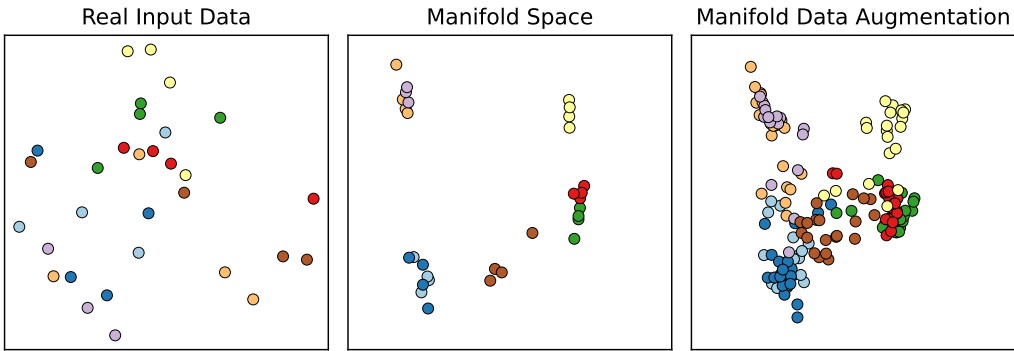

*Figure 4.* PCA linear projection of the first 2 PCs of the raw input data space for the "protein" dataset **(left)**, the manifold space using the encoder from [Hollmann et al., 2023] **(middle)** and the augmented manifold space after using our proposed method **(right)**. The colour of the data points depicts class label.

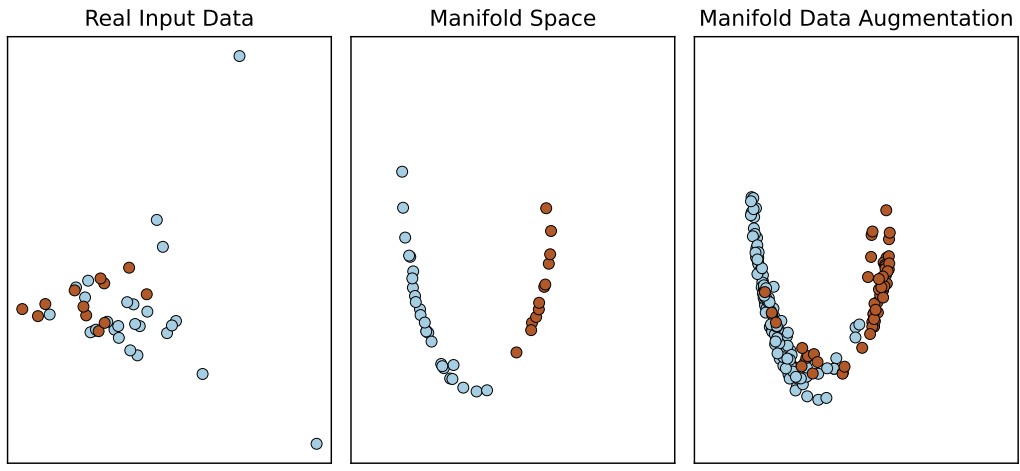

*Figure 5.* PCA linear projection of the first 2 PCs of the raw input data space for the "biodeg" dataset **(left)**, the manifold space using the encoder from [Hollmann et al., 2023] **(middle)** and the augmented manifold space after using our proposed method **(right)**. The colour of the data points depicts class label.

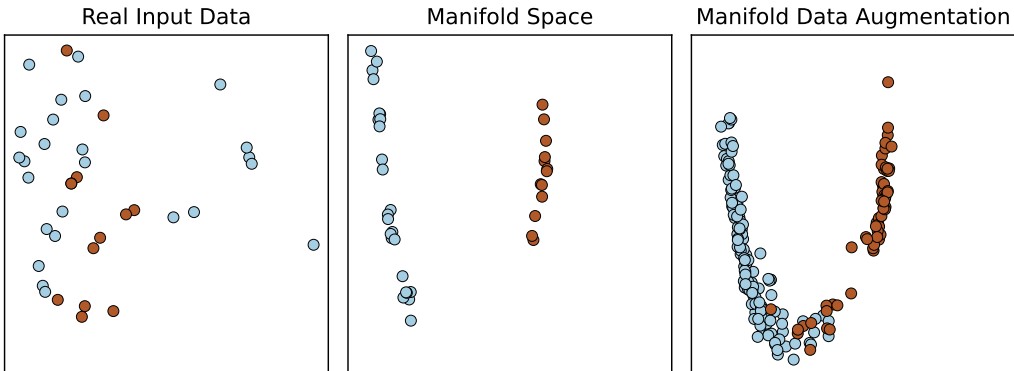

*Figure 6.* PCA linear projection of the first 2 PCs of the raw input data space for the "steel" dataset **(left)**, the manifold space using the encoder from [Hollmann et al., 2023] **(middle)**, and the augmented manifold space after using our proposed method **(right)**. The colour of the data points depicts class label.

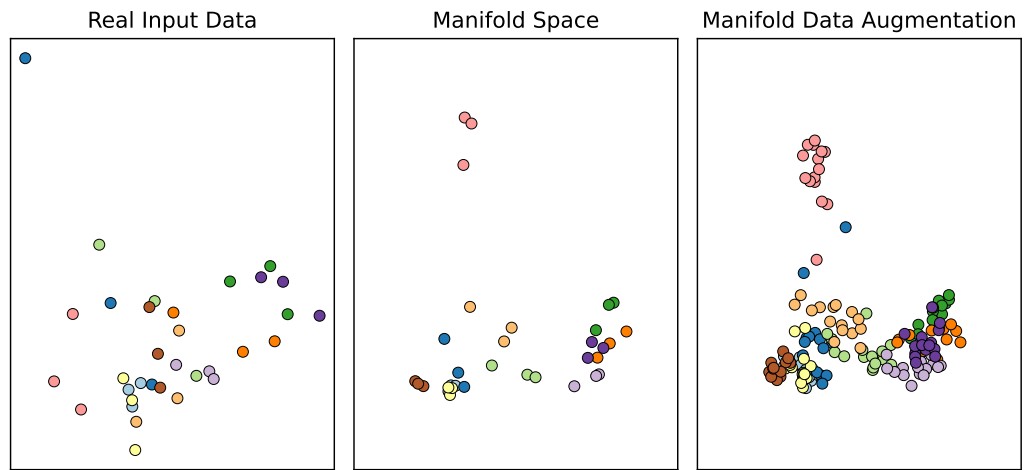

*Figure 7.* PCA linear projection of the first 2 PCs of the raw input data space for the "texture" dataset **(left)**, the manifold space using the encoder from [Hollmann et al., 2023] **(middle)**, and the augmented manifold space after using our proposed method **(right)**. The colour of the data points depicts class label.

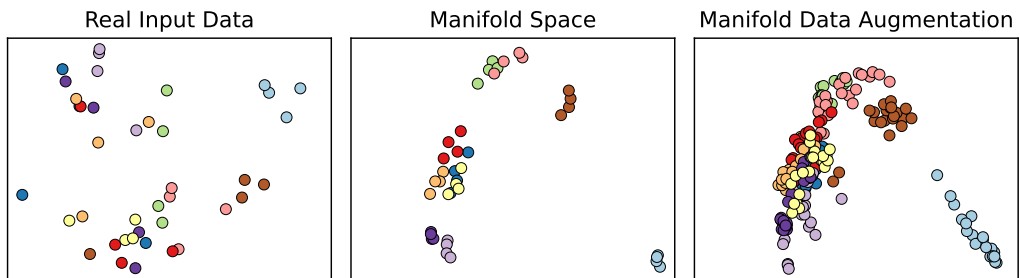

*Figure 8.* PCA linear projection of the first 2 PCs of the raw input data space for the "fourier" dataset **(left)**, the manifold space using the encoder from [Hollmann et al., 2023] **(middle)**, and the augmented manifold space after using our proposed method **(right)**. The colour of the data points depicts class label.

