# OpenReview forum: "TabMDA: Tabular Manifold Data Augmentation for Any Classifier using Transformers with In-context Subsetting"
_ICML.cc/2024/Workshop/ICL — ICML 2024 Workshop ICL Poster_

### Official Review · Reviewer_VDDh · 2024-06-02
**The paper proposes to augment the small tabular datasets by generating embeddings from TabPFN**

**Rating:** 1
**Fit:** 3
**Confidence:** 2

**Workshop Review:**

The author proposes a new way of augmenting small tabular datasets by using "manifold data augmentation". The idea is novel but there are a lot of details that are not clarified properly.

* Clarity and Correctness:
  * the paper is mostly straightforward but there are some key points that are not clear (see below)
  * line 171-173 (algorithm 1): It seems the author is generating the "context dependent" embeddings for X_aug, which is also added to the "context dependent" X_real embeddings. And both "class dependent" versions are fed to the downstream classifiers. However, if this is the case, it is not clear in the writing. In addition, it would need further clarification if this is even considered as data augmentation since the real data is not augmented in the data space but rather the embedding space. And it would need to compare with the non-manifold version if this is the case.
  * The classifier models seem to be untuned

* novelty:
  * The idea of using manifold data augmentation for tabular data is novel but it should be made more clear of what was exactly done as the exact algorithm
  * Missing experimental comparisons. Since this approach is mostly concerned with performance, it would be more convincing to compare TabMDA with at least one other approach such as SMOTE since it's been claimed to beat other approaches https://arxiv.org/abs/2306.15636

Overall, the topic is a clear fit for the workshop. But the main concern from me is the clarify of writing and the lack of baselines and tuning. Therefore I think this paper does not meet the acceptance level of this workshop. However, it is possible that I have missed key information discussed for what is in data space and what is in embedding space. If that is the case, then I think this paper should be accepted.

**Reason For Not Giving Higher Score:**

1. 5 datasets are way too small to test the generalization of the algorithm, especially for hand-picked small tabular datasets.
2. Need to compare to at least one other data augmentation method
3. Need to be more clear about how the real data is encoded and whether if everything is compared in the embedding space or the real data space
4. Some relevant works are missing

**Reason For Not Giving Lower Score:**

1. The idea is novel and helpful for the community
2. The writing is mostly great except some key points (see above)

---

### Official Review · Reviewer_3TK1 · 2024-06-09
**Exploring ICL for Data Augmentation in Tabular Data**

**Rating:** 2
**Fit:** 2
**Confidence:** 3

**Workshop Review:**

Thanks for writing this paper, it looks good.

Here are the problems I see with it that should be addressed:

- Evaluation on only 6 datasets. This is not good practice. You should always evaluate on at least 20 datasets, chosen based on a benchmark, because tabular datasets are super diverse, as you write.

- More of an introduction to what it means to do manifold augmentation would be nice for a general audience as well as an ICL audience.

- It is unclear from where exactly you take the embeddings within TabPFN, please make this clearer. It is also unclear whether you use the TabPFN preprocessing, and if you do, whether you use a fixed seed or whether you also use the randomness of different preprocessings for your model.

- The TabPFN comparison could be improved. First, it is unclear which ensemble setting you use for TabPFN. Second, I could imagine that if you ensemble over random subsets of the original data for TabPFN (as you do for TabMDA), it would yield similar results to your approach, since it is very similar to "bagging".

- Why did you not use TabPFN as the final model as well?

- The paper could be improved by including related literature in stacking, e.g. Autogluon (https://arxiv.org/abs/2003.06505). Which does something similar: it just does not use the embedding, but the prediction of the base models, one of which is TabPFN for Autogluon, as far as I know. Also, they only add the embeddings to examples that are not in the current $X_{ctx}$.


Typos (not part of my review):

- Page 3 "citepPFN"

**Reason For Not Giving Higher Score:**

This paper is a "borderline" paper for me, but I am not given more options. Thus, I think the current rating already is a bit high. I give a list of concrete action points in the top that could improve my rating, though.

**Reason For Not Giving Lower Score:**

This paper seems to establish an interesting idea and has some **very initial** signal that the idea could improve performance, thus I do not want to reject.

---

### Meta-Review · Area_Chair_am5m · 2024-06-12

**Recommendation:** 2

**Metareview:**

The paper introduces TabMDA, a novel method for augmenting tabular data using pre-trained in-context models, specifically leveraging TabPFN. The method maps data into a manifold space and performs label-invariant transformations by encoding data multiple times with varied contexts. This approach aims to improve the performance of machine learning models on small tabular datasets.

The reviews indicate that the paper introduces an interesting and novel idea but falls short in several critical areas, particularly in terms of clarity, breadth of evaluation, and baseline comparisons. The work points towards an interesting direction while more work will be needed in the future.

---

### Decision · Program_Chairs · 2024-06-17

Accept (Poster)